# Volume and Intensity of Stepping Activity and Cardiometabolic Risk Factors in a Multi-ethnic Asian Population

**DOI:** 10.3390/ijerph17030863

**Published:** 2020-01-30

**Authors:** Jennifer Sumner, Léonie Uijtdewilligen, Anne Chu Hin Yee, Sheryl Ng Hui Xian, Tiago V Barreira, Robert Alan Sloan, Rob M Van Dam, Falk Müller-Riemenschneider

**Affiliations:** 1Yong Loo Lin School of Medicine, National University of Singapore, Singapore 119077, Singapore; 2Medical Affairs—Research Innovation & Enterprise, Alexandra Hospital, National University Health System, Singapore 117549, Singapore; 3Department of Health Sciences, University of York, York YO10 5DD, UK; 4Saw Swee Hock School of Public Health, National University of Singapore and National University Health System, Singapore 117549, Singapore; i.uijtdewilligen@amsterdam.nl (L.U.); anne.chu@u.nus.edu.sg (A.C.H.Y.); hui_xian_ng@nuhs.edu.sg (S.N.H.X.); ephrmvd@nus.edu.sg (R.M.V.D.);; 5Department of Exercise Science, School of Education, Syracuse University, Syracuse, NY 13244, USA; tvbarrei@syr.edu; 6Kagoshima University Graduate School of Medical and Dental Sciences, Kagoshima 890-8544, Japan; rsloan@m.kufm.kagoshima-u.ac.jp; 7Institute for Social Medicine, Epidemiology and Health Economics, Charite University Medical Centre, 10117 Berlin, Germany

**Keywords:** physical activity, cardiometabolic risk, step counts, peak cadence

## Abstract

The health benefits of objectively measured physical activity volume versus intensity have rarely been studied, particularly in non-western populations. The aim of this study was to investigate the association between cardiometabolic risk factors and stepping activity including; volume (step count), intensity (cadence) or inactivity (zero-steps/minute/day), in a multi-ethnic Asian population. Participants clinical data was collected at baseline and their physical activity was monitored for seven days, using an accelerometer (Actigraph GT3X+) in 2016. Tertiles (low, moderate, high) of the mean daily step count, peak one-minute, 30-min, 60-min cadences and time/day spent at zero-steps/minute were calculated. Adjusted linear regressions explored the association between stepping activity tertiles and cardiometabolic risk factors. A total of 635 participants (41% male, 67% Chinese, mean age 48.4 years) were included in the analyses. The mean daily step count was 7605 (median daily step count 7310) and 7.8 h of awake time per day were spent inactive (zero-steps/minute). A greater number of associations were found for step intensity than volume. Higher step intensity was associated with reduced body mass index (BMI), waist circumference, blood pressures and higher high-density lipoprotein (HDL). Future health promotion initiatives should consider the greater role of step intensity to reduce cardiometabolic risk.

## 1. Introduction

Non-communicable diseases (NCDs) are responsible for a high degree of morbidity and mortality globally, causing an estimated 38 million deaths per year [1]. Development of an NCD is strongly associated with unhealthy lifestyle behaviors but can be reduced through regular physical activity (PA). Many NCD risk factors, such as high blood pressure, high cholesterol and excess weight, can be reduced by appropriate levels of PA [1,2,3]. International standards i.e. those formulated by the World Health Organization (WHO), recommend that adults should engage in at least 150-minutes (or five times 30-min/week) of moderate-intensity aerobic PA per week [4].

Walking, conceivably the most common form of unstructured PA, can make contributions to PA goals. A survey of 15 European countries reported approximately 37% of the studied population walked 30-min a day five times per week or more [5]. The benefits of walking have been reported in many interventional studies and include: reductions in adiposity, blood pressure and insulin resistance, although improvements in blood lipids appear inconclusive [6,7,8]. Significant associations between walking and reduced cardiovascular disease risk [9,10,11] and increased life expectancy have also been found [12,13]. Walking speed may also influence the degree of effect, with brisk walking rather than slower paced walking reportedly associated with larger impacts on chronic disease risk [14]. 

To date, studies on the effects of objectively measured PA and the associated health effects have largely been conducted in western populations. It is important to understand the PA practices of different ethnic groups, as it cannot be assumed that PA patterns and benefits are the same as those observed in western populations. For instance, the benefits of physical activity are not uniformly understood by all and cultural beliefs can lead to an aversion towards PA [15]. A reluctance to undertake PA is of particular concern in ethnic groups that are pre-disposed to life-style sensitive diseases. For example, there is an increased risk of metabolic disorders, ischemic heart disease and stroke in South-East Asians [16,17]. With greater understanding of PA patterns in different ethnic groups, it is hoped interventions can be tailored more appropriately.

Walking based PA can be considered advantageous over many other forms of PA, in that it is highly accessible requiring no specialist equipment or training and can easily be incorporated into routine life, e.g., as active transport to or from work. As such, public health strategies frequently promote increased walking activity [18,19,20]. Daily step count goals, such as 10,000 steps originating from Japanese walking clubs in the 1960’s [21], are now routinely promoted [18,19,20]. Ten thousand steps can be used as a marker for ‘active’ adults in pedometer-measured activity studies [22]. However, measurement of step volume alone is limited. Step volume does not indicate the intensity of walking and how it contributes towards moderate-intensity activity goals, nor periods of stepping inactivity. To overcome this, research on objectively measured PA is now starting to include cadence i.e. the number of steps taken per minute—a measure of physical intensity [23]. A cadence of ≥100steps/minute has been shown to be equivalent to absolute metabolic equivalent (METS) of ≥ 3 i.e., moderate- to vigorous-PA [24,25]. Assessing cadence in accelerometer measured PA research can therefore provide important insights into patterns of stepping accumulation, step intensity and whether PA goals (i.e., 150-min/week) are met through stepping-based activities. 

Although recent studies have shown beneficial associations between cardiometabolic risk factors, step volume and intensity of stepping [26,27], what remains less understood is whether risk reductions are different for step volume and step intensity. In addition, whether similar patterns of PA and risk reduction are evident in non-western populations. In our previous work we mapped patterns of stepping activity in a multi-ethnic population, but we did not investigate the association with risk factors or whether the effects are different for step volume and step intensity [23]. Therefore, the aims of this study were to investigate: i) the association between cardiometabolic risk factors and step activity, and ii) whether these associations vary by volume (step count), intensity (cadence) or the degree of inactivity (zero-steps/minute/day) in a multi-ethnic Asian population.

## 2. Methods

Recruitment to the study has been described elsewhere [23]. In brief, participants from the Singapore Health Study 2 (SHS2) (*n* = 2686) (a cross-sectional survey of population health) were invited to take part in the accelerometer measured PA study. SHS2 participants were invited to complete the survey by mail and were followed-up by an interviewer at home. Eligible participants had to be permanent residents born between 1933 and 1994. Reasons for exclusion included: pregnancy, severe mental retardation or illness, speech impaired or being bedridden or wheelchair bound. Socio-demographics and clinical history were captured in the survey. Participants were also invited to a clinic-based health screen.

The collected participant information included: age (years), sex (male or female), marital status (married or single category (single category includes single/separated/divorced/widowed)), ethnicity (Chinese, Indian, Malay, other), educational level (low: no formal qualifications/primary school leaving exam/secondary education, medium: 0-levels/ A-levels (equivalent to a high school diploma in the USA) or high: diploma/ university degree or equivalent), employment status (working: currently part or full-time working/student/national service or unemployed: home-maker, retired or unemployed), tobacco use (smoker defined as currently smoking cigarettes or using other tobacco products) and medication use for diabetes, hypertension or lipid management. Clinical measurements, taken during the health screening, were classified according to standardized clinical cut offs in Asia [28,29]: body mass index (overweight ≥ 23 kg/m^2^) derived from height and weight, waist circumference (unhealthy wait size males ≥ 90 cm, females ≥ 80 cm) from the average of the two highest measurements, systolic and diastolic blood pressure (high ≥ 140/90 mmHg) calculated by taking the average of the mean left and right arm measured blood pressure. Fasted cholesterol and diabetic markers were taken once: high-density lipoprotein (HDL, low < 1.0 mmol/L), low-density lipoprotein LDL, high ≥ 3.4 mmol/L), triglyceride (high ≥ 2.3 mmol/L), fasting glucose (high ≥ 6.1 mmol/L) and HbA1c (unhealthy ≥ 6.5%). 

Participants were provided with an accelerometer (ActiGraph GT3X+ ActiGraph corp. Pensacola, FL, USA) and instructed to wear the device continuously for seven consecutive days at the hip and continue their usual routine, except for times when bathing, swimming or sleeping. Although participants could remove the accelerometer prior to bedtime, some participants wore the accelerometers 24-hour a day. To avoid inadvertently identifying sleep time as sedentary behavior, visual inspection of ‘in-bed’ and ’out-of-bed’ times was performed on a minute-by-minute, day-to-day, participant-by-participant basis by two trained researchers. The cleaning protocol to screen for bedtime wearing identified the activity counts which:Began with a period of low activity counts (dropped by ≤ 50% of the participant’s average daytime activity level) to zeros, lasting at least five to ten consecutive min,Activity counts did not rise > 50% of average daytime activity level for more than five consecutive minutes within a three-hour period.

Short periods of activity during sleep periods were permitted if the total duration was < 10 min. Activity for at least ten consecutive minutes was counted as awake time. Data checks were supplemented by information from participant diaries, which recorded bedtimes. 

Raw accelerometer data was extracted and re-integrated in one-minute epochs using ActiLife software™ (version 6, Pensacola, FL, USA). Data were then analyzed using the ‘accelerometery’ package in R [30]. Assessment of the validity of wear time data was according to specifications described elsewhere [31]. Participants with four or more days of wear time for 10 h/day during waking hours were defined as having valid data. 

Cadence data were extracted using the default filter. The following step activity measures were reported: mean daily step count, mean peak one-minute cadence, 30-min cadence and 60-min cadence (cadence times based on existing studies [32,33]) and time spent inactive zero-steps/minute/day (minutes and % of time).

Ethics approval was obtained from the National University of Singapore Institutional Review board (NUS IRB: 13-512). All participants provided written informed consent to take part in the study.

## 3. Statistical Analysis

All analyses were conducted in STATA 14.2 (StataCorp LLC., College Station, TX, USA) and statistical significance was set at *p* < 0.05. Descriptive statistics were calculated as frequencies (%) and mean (with standard deviation). Mean daily step counts were derived by averaging the total steps/day across the days of wear for each participant. The peak one-minute cadence was determined by ranking the individual cadence minutes, identifying the top one-minute cadence and averaging across participants. The peak 30-min and 60-min cadences were determined by ranking the individual cadence minutes, taking the top 30 and 60 minutes respectively (not necessarily consecutive minutes) and averaging across participants. The mean time at zero-steps/minute was calculated by averaging the total time spent at zero-steps/minute/day across the days of wear, subsequently deriving the mean value. To consider whether different levels of step volume and step intensity are associated with risk factors differently, step counts and peak one-minute, 30-min and 60-min cadences were split into tertiles (low, moderate, high activity level), using the lowest activity group as the reference group in each analysis. Time spent at zero-steps/minute was also split into tertiles (low, moderate, high level), using the highest level of non-movement as the reference group in each analysis. Linear regressions, with robust standard errors, were used to explore the association between cardiometabolic risk factors and daily step counts, peak one-minute, 30-min and 60-min cadence and time spent at zero-steps/minutes. The following dependent variables were analyzed as continuous variables: body mass index (BMI, kg/m^2^), waist circumference (cm), systolic (SBP) and diastolic blood pressure (DBP) (mmHg), HDL cholesterol (mmol/L), LDL cholesterol (mmol/L), triglycerides (mmol/L), fasting glucose (mmol/L) and HbA1c (%). All analyses were adjusted for age (years), sex (male/female), ethnicity (Chinese/Malay/Indian/Other), smoking status (smoker yes/no), alcohol consumption (yes/no) and educational level (low/moderate/high). In addition, use of blood pressure medication was included in analyses of blood pressure, use of lipid medications for HDL, LDL and triglycerides, and diagnosis of diabetes for fasting glucose and HbA1c. Analyses of step volume were also adjusted by peak one-minute cadence and cadence analyses were adjusted by daily step volume. Sensitivity analyses were conducted to explore the mediating effect of BMI and the effect of removing rather than adjusting for participants taking blood or lipid medication or those with a diagnosis of diabetes.

## 4. Results

Out of the 895 participants who agreed to take part in the objectively measured PA study, 742 had valid accelerometer data and of those 635 completed the clinic-based health screening and were included in the current analyses (Figure 1). Table 1 presents the characteristics of the sample. Over 60% were overweight ≥ 23kg/m^2^, 45% had an unhealthy waist size, 38% had unhealthy levels of LDL cholesterol and 20% had high blood pressure. Approximately 10% of participants also had high triglycerides, high fasting glucose, high HbA1c and low HDL cholesterol. Age (45.9 v 47.8 years), sex (55% v 58% female), ethnicity (both 66% Chinese), marital status (64% v 61% married) and employment status (74% v 77% working) were not statistically different between those enrolled in the SHS2 and this accelerometer measured PA study, with the expectation of education level, which was higher in the objectively measured PA study.

The mean daily step count was 7605 (Table 1) and the median daily step count was 7310 steps. Approximately half the wear time was spent inactive taking zero-steps/minute (7.8 hours). The mean peak one-minute cadence was 110 (median 113). Mean peak 30-min and 60-min cadences were below moderate-vigorous PA intensity (≥100 steps/minute) at 83 (median 86) and 67 (median 67) respectively. A greater number of risk factors were associated with step intensity rather than step volume (Table 2 and Table 3).

### 4.1. Step Volume

A lower triglyceride level was statistically significant associated with high step volume (highest tertile only). All other analyses were non-significant.

### 4.2. Step Intensity

Lower BMI and waist size readings were consistently significantly associated with peak one-minute, 30-min and 60-min cadences (moderate and high tertiles). The effect was greater in the high compared to moderate cadence tertiles. Lower SBP, lower DBP and higher HDL were generally significantly associated with peak 30-min and 60-min cadences. Lower fasting glucose and Hba1c readings were associated with peak one-minute cadence alone (moderate tertile only). For time spent inactive the only significant association was between the lowest tertile of 0-steps/minute and higher DBP (Table 4). 

### 4.3. Sensitivity Analysis

Full results of the sensitivity analyses are presented in Appendix A. In brief, sensitivity analyses did not alter the findings of the step volume analyses. For step intensity, the inclusion of BMI as a covariate did not change associations with peak 1-min cadence, but for peak 30 and 60-min cadence significant associations with SBP, DBP and HDL were removed. Removing rather than adjusting for participants on blood or lipid medication, cancelled the significant association between peak one-minute cadence and Hba1c, but led to a significant association between Triglycerides and HDL and 30 and 60-min cadences. 

For zero-steps/minute the association to DBP was removed with the addition of BMI as a covariate or when those on blood pressure medications were removed from the analyses. A significant association between fasting glucose, Hba1c and zero-steps minute was gained when those diagnosed as diabetic were removed from the analyses. 

## 5. Discussion

To date, limited research on objectively measured stepping activity and cardiometabolic risk factors has been conducted, particularly on the role of step intensity versus volume in non-western populations. Step volume and intensity were relatively low in this study and the contribution towards conventional PA goals was small. Similar research from the USA has also shown limited contributions towards intensity-based PA goals from stepping activity alone (~7 min at ≥100/steps/min) [34]. Although step activity was low in this study, we identified significant beneficial associations between BMI, waist size and step intensity. Reductions in blood pressure and HDL readings were also associated with higher peak 30 and 60-min cadences to an extent. Conversely, limited associations between risk factors and step volume or time at zero-steps/minute were found. The results indicate that step intensity, rather than volume, is of greater importance in terms of cardiometabolic risk reduction.

It is believed that walking can have many beneficial impacts on health. In a study of ῀9000 adults with impaired glucose tolerance, greater pedometer step activity after 12-months was independently associated with a lower risk of cardiovascular events over five years [35]. Another five-year study reported that increased accelerometer measured step activity, between baseline and after five years, was associated with lower BMI, lower hip-waist ratio and improved insulin sensitivity [36]. Randomized controlled trials of structured walking programs also report benefits to blood pressure, waist circumference, weight and BMI [6,7]. What has been less clear, is whether the effects of stepping are due to step volume or intensity. 

In our study, higher step intensity was associated with a greater number of reductions in risk factors compared to step volume. This fits with studies in Europe that report step intensity, rather than volume of steps, is associated with a greater reduction of cardiometabolic risk factors in adolescents [37] and step intensity reduces the risk of developing metabolic syndrome in adults [38]. The degree of effect is also thought to vary by intensity level. For instance, an observational study found brisk walking rather than slower paced walking had larger impacts on chronic disease risk [14]. In our study, a dose-response effect was observed for certain risk factors. For BMI and waist size, reductions were greater in the high one-minute, 30-min and 60-min cadence quartiles than the moderate quartile. Analysis of randomized controlled trial (RCT) data has been less conclusive on this point, with systematic reviews of walking interventions reporting a lack of evidence demonstrating a dose-response effect [39,40].

Patterns of sedentary behavior were also explored in this study. Sedentary behavior is defined as: “Any waking behavior characterized by an energy expenditure ≤1.5 METs while in a sitting, reclining or lying position” [41]. Systematic reviews on the effects of sedentary behavior report; increased risk of all-cause mortality, cardiovascular-related mortality, cancer-related mortality, increased incidence of cardiovascular disease, cancer and type II diabetes [42,43]. We identified no significant reductions in risk in low or moderate quartiles of inactivity compared to high levels. It has been postulated that the effects of sedentary behavior and physical activity are independent of one another [44]. It is feasible that the degree of sedentary behavior was too low to observe an effect in our study (included adjustment for PA). Another explanation may be the choice of measurement in our study. We used a proxy measure for sedentary behavior; step inactivity i.e., zero-steps/minute/day. A disadvantage of accelerometer measurement is their limited ability to detect specific postures. We were unable to distinguish between periods of standing or sitting, which may lead to misclassification of standing time as sedentary behavior. In addition, periods of prolonged inactivity may have misclassified as sleep time rather than sedentary behavior—participant diaries helped to minimize this risk. Approaches that accurately measure sedentary activities should be used to avoid such limitations. Understanding the type of sedentary behavior is also important as different sedentary activities may not have the same physiological impacts [45]. Future accelerometer studies should collect information on sedentary activities to explore the relationships between sedentary behavior types and health.

In the context of 10,000 steps as marker for ‘active’ adults in pedometer studies [22] and ≥ 100 steps/minute as an indicator of moderate-vigorous physical intensity [24], step activity was suboptimal in this Asian population. Despite a low volume and intensity of step-based PA, associations between cardiometabolic risk and PA were evident, in favor of reducing risk with increasing activity. These findings fit with a growing body of evidence that shows health benefits can be achieved without hitting the conventional 150-min of moderate-intensity aerobic PA per week [4,14,40,46,47]. This has opened up opportunities for more time-efficient PA (e.g., high intensity exercise), which shows similar effects to moderate intensity training [48]. The possibility of lower PA targets, which are beneficial, is encouraging when trying to engage with populations who are less able to exercise, i.e., inactive people, older adults [46,49] or those who are discouraged by the conventional PA target (150-min/week).

### Strengths and Limitations

A major strength of this paper is the use of accelerometers to provide objective measurement of PA levels, which overcomes the bias of self-reported information. However, activities such as cycling, swimming and occupational activities, which do not require stepping, are not accurately captured by accelerometers. Some participants may be undertaking more PA than is reflected by the stepping activity captured. A further strength of this study is the adjustment for confounding, however we cannot rule out residual confounding by unmeasured or imperfectly measured confounders. The main limitation of this study is the cross-sectional design, which means the possibility of reverse causation cannot be excluded. A second limitation is the recruited sample. Although attempts were made to recruit participants from the SHS2, only a proportion consented to our accelerometer-based study. Our sample may be bias, for example only those interested in PA may have taken part. Therefore, the generalizability of our study findings to the wider Asian population in Singapore and beyond may be limited. 

## 6. Conclusions

The results of this study indicate that stepping based PA, particularly high intensity stepping, is associated with lower BMI, waist circumference, blood pressure and increased HDL. Our findings suggest that future health promotion initiatives should place greater emphasis on the role of stepping intensity rather than solely promoting traditional step volume goals, i.e., 10,000 steps. Overall this could help maximize the health benefits from step activity.

## Figures and Tables

**Figure 1 ijerph-17-00863-f001:**
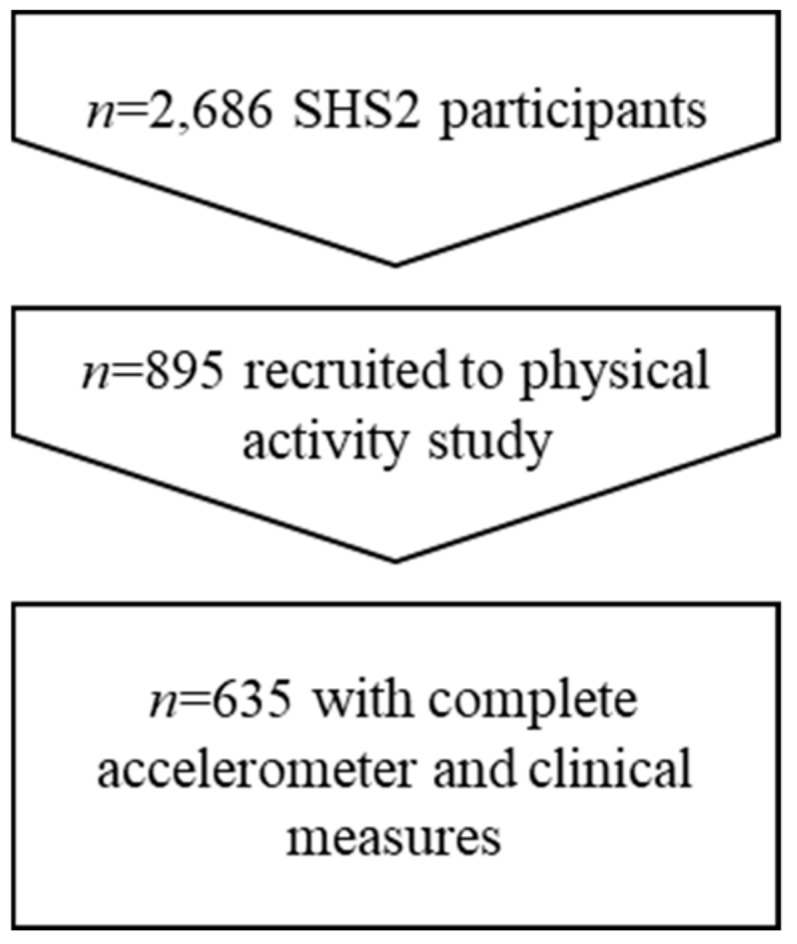
Participant flow.

**Table 1 ijerph-17-00863-t001:** Characteristics of the study population.

Participant characteristics	*N* = 635 *
Mean age years (SD)	48.4 (14.1)
Female (%)	372 (59%)
Ethnicity (%):	
Chinese	425 (67%)
Malay	97 (15%)
Indian	86 (14%)
Other	27 (4%)
Married (%)	395 (62%)
Educational level (%):	
Low	146 (23%)
Medium	190 (30%)
High	299 (47%)
Employed (%)	445 (70%)
Current smoker (%)	82 (13%)
Mean BMI kg/m^2^ (SD)	24.6 (4.5)
Mean waist circumference cm: male (SD) *n* = 263	88.2 (11.1)
Mean waist circumference cm: female (SD) *n* = 372	79.4 (11.0)
Mean systolic blood pressure mmHg (%) *n* = 630	120.7 (17.4)
Mean diastolic blood pressure mmHg (%) *n* = 630	77.6 (10.0)
Mean HDL cholesterol mmol/L (SD): male *n* = 263	1.20 (0.2)
Mean HDL cholesterol mmol/L (SD): female *n* = 370	1.47 (0.3)
Mean LDL cholesterol mmol/L (SD) *n* = 630	3.1 (0.8)
Mean triglyceride mmol/L (SD) *n* = 633	1.3 (0.7)
Mean fasting glucose mmol/L (SD) *n* = 618	5.1 (1.4)
Mean Hba1c % (SD) *n* = 632	5.7 (1.0)
**Accelerometer measures**	
Mean daily wear time minutes (SD)	907.5 (116.2)
Mean daily step count (SD)	7605 (2903)
Mean Peak one-minute cadence steps/min (SD)	110.6 (17.1)
Mean Peak 30-min cadence steps/min (SD)	83.7 (21.3)
Mean Peak 60-min cadence steps/min (SD)	67.4 (20.3)
Mean time at zero-steps/minute minutes (SD)	475.9 (124.4)

SD: standard deviation, BMI: body mass index, HDL: high-density lipoprotein, LDL: low-density lipoprotein; * *n* = 635 unless otherwise stated.

**Table 2 ijerph-17-00863-t002:** Association between step volume and cardiometabolic risk factors.

	Step Activity Tertiles (Mean Daily Steps)		
	Low (4657) Reference Group	Moderate (7380)	High (10,713)
BMI (kg/m^2^)		−0.19 *p* = 0.71	0.35 *p* = 0.51
		(−1.19, 0.81)	(−0.71, 1.42)
Waist (cm)		−0.21 *p* = 0.85	1.54 *p* = 0.23
		(−2.45, 2.02)	(−0.98, 4.06)
Systolic blood pressure (mmHg)		0.67 *p* = 0.66	0.67 *p* = 0.69
		(−2.35, 3.70)	(−2.66, 4.02)
Diastolic blood pressure (mmHg)		0.73 *p* = 0.50	−0.08 *p* = 0.94
		(−1.42, 2.90)	(−2.44, 2.27)
HDL (mmol/L)		0.01 *p* = 0.62	0.04 *p* = 0.18
		(−0.04, 0.07)	(−0.02, 0.10)
LDL (mmol/L)		−0.11 *p* = 0.16	−0.14 *p* = 0.15
		(−0.28, 0.05)	(−0.34, 0.05)
Triglycerides (mmol/L)		−0.04 *p* = 0.60	−0.24 *p* = 0.006
		(−0.20, 0.11)	(−0.41, −0.07)
Fasting glucose (mmol/L)		−0.02 *p* = 0.86	−0.11 *p* = 0.48
		(−0.26, 0.22)	(−0.44, 0.21)
Hba1c (%)		0.05 *p* = 0.49	0.006 *p* = 0.95
		(−0.10, 0.21)	(−0.22, 0.23)

BMI: body mass index, HDL: high−density lipoprotein, LDL: low−density lipoprotein. All multi variable models were adjusted for age, sex, ethnicity, education level, smoking status, alcohol use and peak one−minute cadence. Blood pressure and lipid analyses were also adjusted for medication use. Glucose and hba1c analyses were also adjusted for diagnosis of diabetes.

**Table 3 ijerph-17-00863-t003:** Association between step intensity (cadence) and cardiometabolic risk factors (coefficients, *p* value and 95% confidence intervals).

	Peak One-Min Cadence Tertiles (Mean Cadence)			Peak 30-Min Cadence Tertiles (Mean Cadence)			Peak 60-Min Cadence Tertiles (Mean Cadence)		
	Low (92.3) Reference group	Moderate (113.2)	High (126.6)	Low (59.7) Reference Group	Moderate (86.1)	High (105.8)	Low (44.9) Reference Group	Moderate (68.3)	High (89.7)
BMI (kg/m^2^)		−0.98 *p* = 0.05	−1.84 *p* = 0.001		−1.99 *p* < 0.001	−2.71 *p* < 0.001		−2.00 *p* < 0.001	−2.97 *p* < 0.001
		(−1.97, 0.08)	(−2.98, −0.71)		(−3.01, −0.97)	(−4.00, −1.41)		(−3.05, −0.94)	(−4.39, −1.55)
Waist (cm)		−3.22 *p* = 0.006	−5.37 *p* < 0.001		−3.96 *p* = 0.001	−5.54 *p* < 0.001		−3.92 *p* = 0.002	−6.11 *p* < 0.001
		(−5.51, −0.93)	(−7.99, −2.74)		(−6.38, −1.54)	(−8.62, −2.46)		(−6.41, −1.42)	(−9.43, −2.79)
Systolic blood pressure (mmHg)		−0.70 *p* = 0.63	−2.23 *p* = 0.23		−3.90 *p* = 0.01	−3.06 *p* = 0.13		−3.24 *p* = 0.04	−3.23 *p* = 0.13
		(−3.61, 2.21)	(−5.90, 1.43)		(−6.98, −0.83)	(−7.08, 0.95)		(−6.38, −0.10)	(−7.42, 0.96)
Diastolic blood pressure (mmHg)		−0.22 *p* = 0.83	−2.21 *p* = 0.07		−2.47 *p* = 0.02	−4.46 *p* = 0.001		−1.38 *p* = 0.20	−3.77 *p* = 0.01
		(−2.33, 1.88)	(−4.66, 0.22)		(−4.61, −0.32)	(−7.21, −1.72)		(−3.52, 0.74)	(−6.64, −0.91)
HDL (mmol/L)		0.002 *p* = 0.93	0.06 *p* = 0.07		0.03 *p* = 0.25	0.07 *p* = 0.03		0.04 *p* = 0.13	0.09 *p* = 0.03
		(−0.05, 0.06)	(−0.006, 0.12)		(−0.02, 0.09)	(0.005, 0.15)		(−0.01, 0.10)	(0.008, 0.17)
LDL (mmol/L)		0.07 *p* = 0.37	0.07 *p* = 0.48		−0.01 *p* = 0.84	0.02 *p* = 0.82		0.07 *p* = 0.38	−0.01 *p* = 0.90
		(−0.09, 0.25)	(−0.13, 0.28)		(−0.18, 0.15)	(−0.20, 0.26)		(−0.09, 0.25)	(−0.25, 0.22)
Triglycerides (mmol/L)		−0.06 *p* = 0.42	−0.12 *p* = 0.17		−0.05 *p* = 0.52	−0.13 *p* = 0.18		−0.05 *p* = 0.54	−0.16 *p* = 0.09
		(−0.22, 0.09)	(−0.29, 0.05)		(−0.21, 0.10)	(−0.33, 0.06)		(−0.21, 0.11)	(−0.36, 0.02)
Fasting glucose (mmol/L)		−0.38 *p* = 0.01	−0.24 *p* = 0.16		−0.09 *p* = 0.41	−0.16 *p* = 0.34		−0.16 *p* = 0.17	−0.08 *p* = 0.60
		(−0.67, −0.09)	(−0.59, 0.10)		(−0.33, 0.13)	(−0.50, 0.17)		(−0.40, 0.07)	(−0.41, 0.24)
Hba1c (%)		−0.22 *p* = 0.03	−0.17 *p* = 0.16		−0.001 *p* = 0.98	−0.05 *p* = 0.67		−0.03 *p* = 0.71	0.05 *p* = 0.62
		(−0.43, −0.02)	(−0.42, 0.07)		(−0.15, 0.15)	(−0.28, 0.18)		(−0.19, 0.13)	(−0.15, 0.26)

BMI: body mass index, HDL: high-density lipoprotein, LDL: low-density Lipoprotein. All multi variable models were adjusted for age, sex, ethnicity, education level, smoking status, alcohol use and daily average steps. Blood pressure and lipid analyses were also adjusted for medication use. Glucose and hba1c analyses were also adjusted for diagnosis of diabetes.

**Table 4 ijerph-17-00863-t004:** Association between inactivity (zero-steps/minute/day) and cardiometabolic risk factors (coefficients, *p* value and 95% confidence intervals).

	0-Steps/Minute Tertiles (Mean Daily Minutes at 0-Steps/Minute)		
	Low (347.0)	Moderate (468.5)	High (611.0) Reference Group
BMI (kg/m^2^)	0.54 *p* = 0.24	0.14 *p* = 0.71	
	(−0.37, 1.46)	(−0.64, 0.94)	
Waist (cm)	0.60 *p* = 0.57	0.06 *p* = 0.94	
	(−1.53, 2.74)	(−1.90, 2.03)	
Systolic blood pressure (mmHg)	0.74 *p* = 0.61	0.37 *p* = 0.78	
	(−2.18, 3.67)	(−2.36, 3.10)	
Diastolic blood pressure (mmHg)	2.09 *p* = 0.03	0.61 *p* = 0.52	
	(0.12, 4.06)	(−1.27, 2.49)	
HDL (mmol/L)	−0.02 *p* = 0.47	−0.02 *p* = 0.29	
	(−0.08, 0.03)	(−0.08, 0.02)	
LDL (mmol/L)	0.13 *p* = 0.11	0.07 *p* = 0.33	
	(−0.03, 0.30)	(−0.08, 0.23)	
Triglycerides (mmol/L)	0.01 *p* = 0.81	0.002 *p* = 0.96	
	(−0.12, 0.16)	(−0.13, 0.14)	
Fasting glucose (mmol/L)	−0.07 *p* = 0.65	−0.17 *p* = 0.27	
	(−0.39, 0.24)	(−0.48, 0.13)	
Hba1c (%)	−0.01 *p* = 0.93	−0.12 *p* = 0.18	
	(−0.24, 0.22)	(−0.32, 0.06)	

BMI: body mass index, HDL: high−density lipoprotein, LDL: low−density lipoprotein. All multi variable models were adjusted for age, sex, ethnicity, education level, smoking status, alcohol use and daily average steps. Blood pressure and lipid analyses were also adjusted for medication use. Glucose and hba1c analyses were also adjusted for diagnosis of diabetes.

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
