# Peer review of "Volume and Intensity of Stepping Activity and Cardiometabolic Risk Factors in a Multi-ethnic Asian Population"

_ijerph, 2020, doi:10.3390/ijerph17030863_

Round 1

Reviewer 1 Report

I have no further suggestions to the authors.

Reviewer 2 Report

Authors have addressed all the comments and revised the manuscript accordingly. I have no further comments.

This manuscript is a resubmission of an earlier submission. The following is a list of the peer review reports and author responses from that submission.

Round 1

Reviewer 1 Report

This study aims to understand the association between cardiometabolic risk factors and step activities (including volume, intensity, and degree of inactivity) in a multi-ethnic Asian population. This is an important topic as more and more people are suffering from unhealthy living style by claiming no enough time for exercise. This study will provide evidence on how step activities can benefit health.

Major comments:

Page 3, line 138: measure the top 30- and 60-minute steps/minute in consecutive minutes makes more sense to me. Please elaborate your theory on how inconsecutive minutes has the same effect on patients’ health as measuring top 30- and 60-minute steps/minute in consecutive minutes. Table 1 on page 4: are there any patients who were underweight? How did you deal those patients in the model? Table 1 on page 5: need to provide more explanation on why total number of patients are different on outcome measures (SBP, DBP, HDL among female, LDL, triglyceride, fasting glucose, and HbA1c)? How did you deal with this in the analysis? Discussion section: the authors should link the previous literature more closely with the current study by comparing your results vs. others. Moreover, it seems there’s lacking discussion on inactivity vs. cardiometabolic risk factors. The last paragraph of Discussion section on page 9 started with strength but end with limitation without clear transition. Should have stated strengths and limitations more clearly. Content fluency and punctuations throughout the paper needs major improvement.

Minor comments:

Page 2, line 91: did you not include single patients in the single category? page 2, line 93: what do you mean by 0-levels/A-levels in medium educational level? Please elaborate. Page 3, line 130-131: this has already been stated on page 3, line 103-105. Page 8, line 212-213: the reference group is low step intensities, so the sentence “The effects were larger in higher step intensities compared to moderate step intensities” need to be revised and discuss in a clearer way. How the citation cites in the paper should be consistent. For instance, there should be no space between the citation and the content, some citations are after the period, but some are before the period, etc.

Author Response

1.     Page 3, line 138: measure the top 30- and 60-minute steps/minute in consecutive minutes makes more sense to me. Please elaborate your theory on how inconsecutive minutes has the same effect on patients’ health as measuring top 30- and 60-minute steps/minute in consecutive minutes.

Many studies have explored whether health benefits are different between bouted (usually >10mins of consecutive activity) and non-bouted physical activity (examples below). Generally, the effects are similar, therefore it can be appropriate to look at non-bouted activity.

·       Effects of bouted versus non-bouted minutes of activity on mortality. “Mortality risk reductions associated with MVPA are independent of how activity is accumulated”.

doi: 10.1161/JAHA.117.007678.

·       Effects of bouted versus non-bouted MVPA on HRQOL. Similar effects from bouted and non bouted MVPA on HRQOL were observed.

doi.org/10.1016/j.pmedr.2015.12.005 2211-3355

·       Associations between bouted v non-bouted activity and various biologically measures e.g. triglycerides “results were similar for both approaches.”

DOI: 10.4278/ajhp.110916-QUAN-348

2.     Table 1 on page 4: are there any patients who were underweight? How did you deal those patients in the model?

Some of the participants were considered underweight according to their BMI. In the analysis we included BMI as a continuous variable not categorical.

3.     Table 1 on page 5: need to provide more explanation on why total number of patients are different on outcome measures (SBP, DBP, HDL among female, LDL, triglyceride, fasting glucose, and HbA1c)? How did you deal with this in the analysis?

The difference in total number for specific variables is due to missing data-this is clearly identified in Table 1. The amount of missing data was minimal and we do believe this would of lead to a substantial impact on study. The analysis was performed on participants with complete data only.

4.     Discussion section: the authors should link the previous literature more closely with the current study by comparing your results vs. others. Moreover, it seems there’s lacking discussion on inactivity vs. cardiometabolic risk factors.

We have revised the discussion to include content on our inactivity findings and added further comparisons to previous literature.

5.     The last paragraph of Discussion section on page 9 started with strength but end with limitation without clear transition. Should have stated strengths and limitations more clearly.

We have added a sub header “5.1 strengths and limitations” to more clearly identity this part of the discussion. The text has also been revised for clarity.

6.     Page 2, line 91: did you not include single patients in the single category?

Yes, the text has been amended to make this clearer.

7.     page 2, line 93: what do you mean by 0-levels/A-levels in medium educational level? Please elaborate.

We have attempted to categorise the educational level of the cohort into three groups low, medium, high. O/A-level education is classed as a ‘medium’ level of education. A-level is equivalent to a high school diploma in the US. We have amended the text to reflect this so it is clearer for the readership.

8.     Page 3, line 130-131: this has already been stated on page 3, line 103-105.

Duplication removed.

9.     Page 8, line 212-213: the reference group is low step intensities, so the sentence “The effects were larger in higher step intensities compared to moderate step intensities” need to be revised and discuss in a clearer way.

We have revised the text so it is clear the comparison group is the low quartile.

10.  How the citation cites in the paper should be consistent. For instance, there should be no space between the citation and the content, some citations are after the period, but some are before the period, etc.

Revised.

Reviewer 2 Report

This study examined whether the association between stepping volume (daily step count) and cardiometabolic risk factors is different from the association between stepping intensity (cadence) and cardiometabolic risk factors in Asian population. The results showed that the association with cardiometabolic risk factors are greater for stepping intensity than stepping volume. This finding is novel for Asian population and is clinically relevant. This study was well-designed and controlled carefully. The only one suggestion is that please add unit to the “mean wear-time” in the Table 1.

Author Response

1.     please add unit to the “mean wear-time” in the Table 1.

Added.

Reviewer 3 Report

This study was well conducted, and the data were clearly presented. I have some comments for authors to clarify further.  

Please further clarify the indication of peak 1-minute, 30-minutes, and 60 minutes cadences for step intensity. Do they represent different level/strength of intensity and whether they were all highly correlated? It is also important to show the correlations between step volume, step intensity, and inactive time. Line 27, how does the median involve in the tertile calculation? Line 172 states that the median daily step count was 7310 steps (Table 1), which is not the data presented in Table 1, i.e. mean daily step count of 7605 steps. The following sentences also state the median of inactive time, peak 1, 30, 60-minutes cadences, rather than the mean presented in Table 1. If the cardiometabolic and accelerometer measures were not normally distributed, median (interquartile range) should be presented in Table 1 rather than mean (SD). Please add the unit for the accelerometer measures in Table 1, i.e. mins for wear-time, steps/day for daily step count, steps/min for peak 1, 30, 60-minute cadence. Also, % for mean Hba1c. Please discuss what could be the underlying mechanisms explaining step intensity and reduced BMI, WC, blood pressure and increased HDL. Evidence has shown a link between sedentary behaviour and increased risk of cardiometabolic diseases, but this study found no evident association between inactive time and cardiometabolic risk factors. Please further discuss this finding.  

Author Response

1.     Please further clarify the indication of peak 1-minute, 30-minutes, and 60-minutes cadences for step intensity. Do they represent different level/strength of intensity and whether they were all highly correlated?

Peak 1, 30 minute and 60-minute cadences were developed to measure the intensity of physical activity. They are indicators of how active (in terms of intensity) an individual is.   

Peak 1-minute cadence represents the highest intensity level achieved.

Peak 30-minute cadence is an indicator for the natural best 30-minutes of effort in a day.

Peak 60-minute cadence is an indicator for the natural best 60-minutes of effort in a day.

We did not feel it was necessary to explore correlation between these measures. However, we have previously look at the correlation between step volume and intensity (cadence) (doi:10.1186/s12889-018-5457-y).

2.     It is also important to show the correlations between step volume, step intensity, and inactive time.

In our previous article (doi:10.1186/s12889-018-5457-y.) we explored the correlation between daily step count and peak cadences and found a high degree of correlation. We did not wish to repeat the analysis for this publication.

3.     Line 27, how does the median involve in the tertile calculation?

This should be mean, corrected.

4.     Line 172 states that the median daily step count was 7310 steps (Table 1), which is not the data presented in Table 1, i.e. mean daily step count of 7605 steps.

We have revised to report both mean and median in the free text.

5.     The following sentences also state the median of inactive time, peak 1, 30, 60-minutes cadences, rather than the mean presented in Table 1.

We have revised to report both mean and median in the free text.

6.     If the cardiometabolic and accelerometer measures were not normally distributed, median (interquartile range) should be presented in Table 1 rather than mean (SD).

Whilst in principal we agree with this comment we have observed that a great proportion of accelerometer studies report means-despite skew. So our results were easily comparable to the existing evidence base we decided to report means. However, we have included medians in the free text for some of the key step descriptors in this study.

7.     Please add the unit for the accelerometer measures in Table 1, i.e. mins for wear-time, steps/day for daily step count, steps/min for peak 1, 30, 60-minute cadence. Also, % for mean Hba1c.

Added.

8.     Please discuss what could be the underlying mechanisms explaining step intensity and reduced BMI, WC, blood pressure and increased HDL.

The purpose of this paper was to investigate the associations between cardiometabolic risk factors and step activity and how these vary by volume, intensity or the degree of inactivity. Whilst we appreciate the suggestion we do not feel discussion on the physiology mechanisms is the focus of this paper. However, we have substantially revised the discussion section based on your and other reviewer comments.

9.     Evidence has shown a link between sedentary behaviour and increased risk of cardiometabolic diseases, but this study found no evident association between inactive time and cardiometabolic risk factors. Please further discuss this finding.

We have included some extra discussion on this point towards the end of the discussion section.

Reviewer 4 Report

This is an interesting piece of research. Main strengths are the sample size and having assessed PA objectively.

There is a strong limiation, since the participants were not asked to refrain from any sport activity that was not registered by the accelerometers.

My advices are mainly two:

A more in deep justification of why ethnic is a factor that can influence PA behaviour. A more in deep discussion of why intensity seems to have a deeper impact on health than volume is welcomed.

Author Response

1.     A more in deep justification of why ethnic is a factor that can influence PA behaviour.

The introduction has been revised to include more information/justification on the role of ethnicity.

2.     A more in deep discussion of why intensity seems to have a deeper impact on health than volume is welcomed.

We have revised the discussion to include greater discussion on intensity versus volume.

Reviewer 5 Report

Issues
Introduction
1. The introduction is lacking to provide the aim of this study.

2. Did the authors explain the define of cardiometabolic risk factors correctly? Are there cardiometabolic biomarkers? Isn't that a risk factor for metabolic syndrome?

3. Is the hypothesis set up adequately other than the lack of explanation for the purpose?

Methods
4. Please add the IRB approval number.

5. Please add a chart of the criteria for selecting and excluding the study participants as a figure.

6. Line 96-103, add a reference to Asian clinical measurement standards.

Statistical analysis
7. Please add a description of the five primary considerations in the correlation analysis.

Result
8. The results of this study showed no significant differences in overall data.
Have you processed the statistics separately for men and women?

9. There is no cardio-metabolic risk factor, and it is considered a risk factor for metabolic syndrome. Please correct the word as a risk factor for metabolic syndrome.

Discussion
10. In the discussion section, please add a paragraph about the limitations of the study.

11. No significant difference appeared in the overall variable. Please more deeply explain it.

12. Is there a sufficient explanation of similarity and differentiation in comparison with the previous prior study?

Author Response

1.     The introduction is lacking to provide the aim of this study.

The introduction has been revised to provide a clearer justification for the project.

2.     Did the authors explain the define of cardiometabolic risk factors correctly? Are there cardiometabolic biomarkers? Isn't that a risk factor for metabolic syndrome?

We used the cardiometabolic risk factors to be consistent with previously conducted research in this field. The NHANES study – a large US based accelerometer and health study looked at the relationship between “step-based accelerometer metrics and cardiometabolic risk factors” including waist circumference, BMI, BP, glucose and cholesterol.

doi: 10.1249/MSS.0000000000001100

We confirm we did not look at biomarkers in this study.

3.     Is the hypothesis set up adequately other than the lack of explanation for the purpose?

We did not include a hypothesis. The aims of the study are set out in the last paragraph of the introduction.

4.     Please add the IRB approval number.

Added. NUS IRB: 13-512.

5.     Please add a chart of the criteria for selecting and excluding the study participants as a figure.

Added.

6.     Line 96-103, add a reference to Asian clinical measurement standards.

Added.

7.     Please add a description of the five primary considerations in the correlation analysis.

To clarify we did not conduct a correlation analysis. In this study we performed a multi-variate linear regression analysis. All models used robust standard errors to account for any data normality issues. We do not believe it is standard convention to include a description of such data normality checks and have not amended the text.

8.     The results of this study showed no significant differences in overall data. Have you processed the statistics separately for men and women?

No, we did not do an analysis by sex, although sex was included as a covariate in all regressions. Due to the sample size we were unable to perform further sub-group analysis.

9.     There is no cardio-metabolic risk factor, and it is considered a risk factor for metabolic syndrome. Please correct the word as a risk factor for metabolic syndrome.

This line refers to two different studies. We have revised the text to clarify this sentence.

10.  In the discussion section, please add a paragraph about the limitations of the study.

We have added a sub-header in the discussion section “strengths and limitations” to more clearly identify this part of the discussion.

11.  No significant difference appeared in the overall variable. Please more deeply explain it.

We are unclear as to what the reviewer means by this point and are therefore unable to respond to this comment.

12.  Is there a sufficient explanation of similarity and differentiation in comparison with the previous prior study?

The introduction (last paragraph) has been amended to clearly distinguish between these two studies.